# Age-Related Changes in the Fibroblastic Differon of the Dermis: Role in Skin Aging

**DOI:** 10.3390/ijms23116135

**Published:** 2022-05-30

**Authors:** Alla Zorina, Vadim Zorin, Dmitry Kudlay, Pavel Kopnin

**Affiliations:** 1Human Stem Cells Institute, 119333 Moscow, Russia; doc_zorin@mail.ru (A.Z.); zorin@hsci.ru (V.Z.); 2Department of Pharmacology, Institute of Pharmacy, I. M. Sechenov First Moscow State Medical University (Sechenov University), 119991 Moscow, Russia; d624254@gmail.com; 3N. N. Blokhin National Medical Research Oncology Center, Ministry of Health of Russia, 115478 Moscow, Russia

**Keywords:** skin, aging, fibroblasts, stem cells

## Abstract

Skin aging is a multi-factorial process that affects nearly every aspect of skin biology and function. The processes developing in the skin during aging are based on fundamental molecular mechanisms associated with fibroblasts, the main cellular population of the dermis. It has been revealed that the amount of fibroblasts decreases markedly with age and their functional activity is also reduced. This inevitably leads to a decrease in the regenerative abilities of the skin and the progression of its aging. In this review we consider the mechanisms underlying these processes, mainly the changes observed with age in the stem/progenitor cells that constitute the fibroblastic differon of the dermis and form their microenvironment (niches). These changes lead to the depletion of stem cells, which, in turn, leads to a decrease in the number of differentiated (mature) dermal fibroblasts responsible for the production of the dermal extracellular matrix and its remodeling. We also describe in detail DNA damages, their cellular and systemic consequences, molecular mechanisms of DNA damage response, and also the role of fibroblast senescence in skin aging.

## 1. Quantitative Changes in the Population of Dermal Fibroblasts

With age, a decrease in the number of cells is observed in the population of dermal fibroblasts (DFs), which has been confirmed in a number of scientific papers. Varani et al. (2006) examined skin biopsies of young people (18–29 years old) and older people (80 years and older) isolated from areas of the body protected from ultraviolet radiation (UV). It has been shown that the total number of DFs in the older group was reduced by an average of 35% [1]. A study by Solé-Boldo et al. (2020), which was conducted using single-cell RNA sequencing (scRNA-seq) of DFs obtained from the photoprotected skin of young and elderly people (transcriptomes of more than 5000 DFs were analyzed), confirmed a decrease in the number of these cells with age. At the same time, lowering their heterogeneity was also revealed [2]. The decrease in the number of DFs has a significant impact on the biology of the skin since fibroblasts are the leading cells of the dermis, which is the connective tissue basis of the skin. They are responsible for the production, organization, and renovation of the components of the extracellular matrix (ECM) that determines the morphofunctional characteristics of the connective tissue [3,4,5,6,7].

Thus, a reduction in the amount of DFs leads to skin homeostasis disturbance and is accompanied by significant changes in the skin micro- and macrostructure [8]. These processes in the DF populations have a selective pattern. In particular, a study by Mine et al. (2008), analyzing the characteristics of papillary and reticular fibroblast populations of the dermis, showed that, with age, the most prominent changes are observed in the fibroblast population of the papillary layer [9]. This is probably due to the predominance of stem/progenitor cells (SCs, which are tissue-specific mesenchymal cells, precursors of fibroblasts of the skin outside the hair follicles and mesenchymal stem cells, such as bone marrow stem cells and others that were not incorporated into the skin during embryonic development) in the composition of the papillary fibroblast populations in which changes primarily occur [10].

## 2. Age-Related Changes in the Dermal Stem Cell Population

The main feature of SCs is the ability to self-renew and differentiate. By doing so, SCs play a key role in maintaining the tissue homeostasis providing the skin (as well as the other body tissues) with differentiated mature descendant cells [11,12,13,14]. It has been revealed that there is a decrease in the quantity and quality of the SCs functions with age [15,16], which leads to depletion of the SCs pool and, thus, to a decrease in the regenerative abilities of body tissues, one of the main causes of tissue aging [15,17,18,19,20,21,22,23].

### 2.1. Depletion of the SCs Pool

The depletion of the SCs pool can be caused by the whole range of factors, the main of which are the following:A decrease in the ability of SCs to self-renew and the lowered proliferative potential [12,15,24], in particular, are due to a reduction in the number of receptors for growth factors located on the SCs’ surface membranes [25]. As a result, there is a decrease in the number of SCs capable of responding to signals stimulating their proliferation [26].Excessive SC proliferation causing the depletion of the cell niche [18,19]. It has been revealed that the SCs’ regenerative potential is limited by a certain number of cell divisions during the life of the organism [27]; therefore, the prolonged activation of a stress factor stimulating the proliferation of SCs inevitably leads to depletion of their pool or to aberrant differentiation of these cells [28,29].Increased apoptosis [22,30].Telomere shortening [18,31].Cellular aging [22,24,31,32].

Damages in the SCs themselves (and their accumulation with age) affect their DNA, mitochondrial (Mt) and protein complexes [11,18,33,34], epigenetic regulation of the genome (histone modification, DNA methylation, etc.), and other cellular structures [13]. The results of numerous studies indicate that the changes associated with the depletion of the SCs pool may be associated not only with disorders in the SCs themselves, but also with disorders in their niches (microenvironment) from which the cells constantly receive signals ensuring their proper functioning [21,35,36,37,38]. Thus, it has been revealed that disturbance of the ECM’s mechanical properties with age—in particular, an increase in its rigidity—can impair the SCs’ functions [39,40,41].

Studies by Sun et al. (2011) and Shakouri-Motlagh et al. (2017) showed that cultivation of MSCs (mesenchymal stem cells) isolated from “old” tissues on a native substrate significantly increases the ability of these cells to proliferate and differentiate [39,40] and, therefore, enhances their expansion. Koester et al. (2021) investigated the fundamental mechanisms of SCs aging by determining the age-dependent changes in the chromatin landscape of SCs isolated from mouse hair follicles (HFSCs). The results of the study also confirmed the significance of the “mechanical” state of the HFSCs niche that, in particular, affects the availability of chromatin and influences the epigenetic status of the key genes responsible for SCs’ self-renewing and differentiation [41]. It has been revealed that the ECM stiffness (in this study, the basement membrane) observed with age leads to a decrease in the availability of HFSCs chromatin. According to the author’s opinion, the rigid matrix of the SCs niche promotes the development of mechanosensory stress, which induces epigenetic changes in the genome that cause an impairment of the SCs’ functions.

Thus, the depletion of the SCs pool is a collective consequence of the many damages observed in the cells and in their microenvironment. However, it mainly correlates with disorders in the genetic apparatus of the cell [19], which lead to instability of the genome and finally terminating the proliferative life of the cell [36].

### 2.2. Changes in the SC Genome

Disorders in the genetic apparatus of the cell occur under the influence of a number of factors, both internal and external. The internal ones include [14,18]:Reactive oxygen species (ROS) produced by metabolic intermediates and dysfunctional Mt;DNA replication and DNA repair errors;Glycation end products;Dysfunction and shortening of telomeres (the end regions of chromosomes that ensure stable cell replication and protect chromosomes from fusion);Inflammation of the surrounding tissue.

As a result of these factors action, about 100,000 DNA damages can occur per day in each cell, including various types of nucleotide modifications and single- and double-stranded breaks [14,42]. The second group of external factors includes UV and X-rays, chemicals, tobacco smoke, alcohol, and viral infections [12,36,43,44].

For skin cells, the most significant is the effect of ROS that promote the induction of both chronological skin aging and photoaging by damaging the nucleus and other structures/organelles of the cell [44,45,46,47,48]. During photoaging, ROS (superoxide anion (O_2_^−^), hydrogen peroxide (H_2_O_2_), etc.) are formed under the influence of UV light as a result of photochemical reactions. During chronological skin aging, ROS generation regularly occurs in DFs and keratinocytes from molecular oxygen in the process of aerobic respiration and cellular metabolism [49].

The decrease in reparative abilities of the SC genome observed with age also plays a significant role in its instability. This process leads both to the impossibility to repair the occurring damages and to the emergence of errors in reparation, which cause new genomic damages and progressive accumulation of mutations [18,50,51]. A study by Garagnani et al. (2021), using the whole genome sequencing of centenarians (with average age of 106.6 years), confirmed that longevity is connected to a certain genetic pattern associated with effective DNA repair mechanisms and a low level of somatic mutations. According to the author’s opinion, this is the key factor of cellular homeostasis [52].

Changes in the SC genome appear to be the result of the realization of a genetic program that must ensure the genomic stability and the dynamics of expression of various genes at different stages of ontogenesis [5,53]. It has been shown that the aging process in humans is controlled by evolutionarily conservative genetic signaling pathways [18]. Genetic and biochemical analysis of SCs in various tissues has revealed age-related changes in DNA methylation, histone acetylation, and methylation markers. The research results suggest that epigenetic regulation is also the evolutionarily conservative mechanism involved in SCs aging in the human tissues [20,54,55].

## 3. Cellular Aging or Senescence

The state of genetic instability accompanied by a violation of the transcription process of vital genes and destroyed cellular metabolism leads to cellular aging [56], which is phenotypically associated with a wide range of changes at the molecular and cellular levels, including genomic (double-stranded DNA breaks, telomere shortening, decreased efficiency of repair processes) and epigenomic changes, Mt dysfunction, impairment of proteostasis and functions of other cellular structures, and deregulation of signaling pathways [12,13,36,57,58] (Figure 1).

Thus, DNA damage causes a response from the cell (DNA damage response, DDR), which is generally aimed at preventing the proliferation of damaged cells that are potentially oncogenic [18,31,59]. DDR involves a coordinated cascade of autonomous cellular and non-cellular reactions, including various pathways of DNA repair, cell apoptosis, and control points of the cell cycle, and its irreversible stop, causing cellular aging or senescence (Lat. *senex*, “old”) that, apparently, makes the most significant contribution to depletion of the SC pool [19,22,24,31,32,36,44,48,60,61,62] (Figure 2).

At the molecular level, the process of cellular aging is associated with the following mechanisms. The persistent damage to nuclear DNA triggers a activation of DNA damage response (DDR) pathways, which activates the signaling pathways of aging effectors - inhibitors of cyclin-dependent kinase p53-p21 and/or p16INK4a [63,64] (Figure 2) [36,65,66]. The p53-p21 signaling pathway is activated first, which initiates a temporary stop of the cell cycle, thereby preventing the transmission of damaged genetic information [63,64,65,66]. Then, if the damage is minor and subject to repair (it should be noted that most DNA damages can be corrected in the course of reparation), the cell cycle is restarted. In the case when the DNA damage cannot be corrected, the cell undergoes apoptosis. At the same time, the p21 signaling pathway, by producing the chemokine CXCL14, activates the genes of immune “surveillance”, attracting macrophages that control cells exposed to stress. This provides the first line of defense against dysfunctional cells. If the damaged cells are restored within four days, the macrophages abandon their targets. Otherwise, macrophages mobilize cytotoxic T lymphocytes, which promote the elimination of damaged cells [67]. 

In the case of persistent DNA damages (and/or their accumulation), an irreversible arrest of the cell cycle occurs, and the cell acquires a senescent phenotype (Figure 2) [19,36,62,68]. It has been determined that in response to cellular stress, protein p21 changes the regulatory features of pRb transcription (pRb is a retinoblastoma protein, a suppressor of carcinogenesis) and leads to inhibition of the cell cycle genes [36,69], while the protein p16 INK4a plays the key role and regulates the long-term cell cycle arrest via the pRb-E2F signaling pathway (pRb is the main mediator supporting the irreversibility of the cell cycle) [36,69]. With age, this protein accumulates significantly in tissues [66,70,71]).

Cellular senescence was first described in the 1960s, when L. Hayflick and P.S. Moorhead (1961) established that human fibroblasts in culture undergo a certain number of divisions (about 50) before cell expansion is stopped [72]. This limit of cell divisions, called the Hayflick limit, is a result of progressive shortening of telomeres with each cell division. This is a physiological reaction aimed at preventing the accumulation of DNA damages [73]. When telomere shortening reaches a critical level, DDR occurs, finally leading to either cell apoptosis or irreversible cell cycle arrest. One of the main causes of telomere shortening is a decrease in the level/activity of enzyme telomerase, which is observed with age [74]. This type of cellular aging is called replicative aging [62].

Along with replicative aging, all human diploid cells, including DFs, can undergo premature aging unrelated to telomere shortening [62,65]. The premature cellular aging can be divided into two types depending on the DNA damage triggers: (1) oncogen-induced cellular aging, developing in response to oncogen activation; and (2) stress-induced cellular aging, resulting from oxidative stress under the influence of ultraviolet light, ionizing radiation, and ROS [44,48,62,65,75]. For example, it has been shown using a DFs culture (in vitro) that tobacco smoke components cause oxidative DNA damage, leading to premature cell aging and further to irreversible cell cycle arrest [18,76].

It should be noted that regardless of the trigger that induces cellular aging and regardless of the type of cellular aging, the irreversible stop of the cell cycle is general in the process of cellular aging [74,75], as well as the acquisition by cells of the senescent phenotype (senCs, senDFs) (Figure 3).

The damage of the cell nucleus inevitably leads to impairment of other cellular structures/organelles, which is accompanied by an increase in degenerative processes. Thus, the increasing instability of the nuclear genome observed with age is critical for the functioning of mitochondria (Mt), which are the key organelles in eukaryotic cells that play a central role in ATP production and cell bioenergetics [56,77,78,79,80,81]. Many proteins necessary for Mt activity are encoded by nuclear genes and Mt dysfunction occurs when nuclear genes are damaged [80,82]. In addition, when DDR is activated, a decrease in the amount of Mt occurs, as well as exhaustion of their membrane potential and impairment in the electron transport chain and the mitophagy process [83,84]. As a result of Mt changes, the production of ROS increases in cells, which promotes the progressive damage of proteins and lipids, as well as the accumulation of lipofuscin (an insoluble covalently crosslinked glycolipoprotein complex) [85], telomere dysfunction [86], and degradation of lysosomes. All these processes cause an even greater increase in ROS production. Thus, a vicious autocrine feedback circle is initiated that increases the ROS-mediated damage to nuclear DNA, which leads to the progression of SCs aging [44,45,87,88,89,90,91,92].

It should be noted that the accumulation of lipofuscin underlying age-related skin pigmentation disorders is also promoted by the decrease in autophagy observed with age, which is characteristic of senDFs [93]. According to research results, Mt dysfunction serves as an important factor inducing both chronological aging and photo-aging of the skin, possibly creating a common link between them [44,94,95]. In the case of both types of skin aging, the high level of mutations (deletion (decay) of 4977 base pairs) has been revealed in mitochondrial DNA of DFs [96].

In the process of cellular aging, the disruption of proteostasis is also observed, which is manifested by the accumulation of modified and improperly folded proteins and protein aggregates, which occurs due to a decrease in the ability of cellular proteasomes to eliminate the protein aggregates [97,98,99]. This leads to a decrease in cellular functions, as well as to oxidation of the cellular membrane lipids, which promotes a reduction in the efficiency of transmembrane transport and a disruption in the transmembrane signaling pathways [24].

The abovementioned studies confirm the conclusion that cellular aging is a complex process involving structural damage and impairment of the vital activity of almost all cell organelles. As a consequence, an autocrine vicious circle is formed that violates cell functioning.

## 4. Senescent Fibroblasts and “Paracrine” Skin Aging

Senescent DFs (senDFs) like other senescent cells (senCs) undergo morphological changes, promoting their resistance to apoptosis, which contributes to the accumulation of these cells in the dermis [62,100] and changes the pattern of gene expression. The resistance of senDFs to apoptosis may be caused by an increase in the levels of anti-apoptotic proteins such as BCL-2 and FOXO4 [101,102]. It may also be associated with the secretion of non-classical MHC HLA-E, which activates inhibitory receptor NKG2A on natural killer cells (NK cells) and CD8+ T lymphocytes. Activation of NKG2A prevents the elimination of senDFs [103]. The altered pattern of gene expression leads to reprogramming of senCs metabolism and secretion of a wide range of soluble and insoluble factors which are commonly referred to secretory phenotype associated with aging (SASP, Senescence-Associated Secretory Phenotype) [104,105]. SASP includes many pro-inflammatory cytokines/chemokines, growth factors, metalloproteinases (MMPs), and other soluble factors (Figure 4) [46,106]. Due to secretion of these factors, cells with SASP by a “paracrine” way promote the aging of neighboring SCs, inflammation of the tissue, and destruction/degradation of dermal ECM through the action of MMPs. In addition to the soluble SASP factors, extracellular vesicles (exosomes) produced by senCs are also important mediators of the “paracrine” aging. By means of their constituent microRNAs, specific proteins, nucleic acids, and other insoluble factors, senCs perform intercellular connections, in particular between senDFs and neighboring SCs, enhancing the aging processes in these cells (as has been shown both in vitro and in vivo) [107].

Basisty et al. (2020) published the senDFs SASP atlas, which contains a large database, including the soluble factors and exosomes produced by senDFs. The authors revealed that SASP is variable and its composition may change depending on the trigger factor that causes cell aging [108]. Waldera Lupa et al. (2015), by examining the DFs obtained from the skin of elderly people with chronological aging, revealed a specific secretome, including 70 proteins corresponding to the classic SASP and 21 unique proteins that, according to the authors, may reflect specific skin-aging processes [109].

### 4.1. Identification of Senescent Fibroblasts

Senescent DFs (like other senCs) are cells that do not have specific markers; so, they can be identified by means of a combination of characteristic features [76]. The following senDFs markers are most frequently used in vitro (they were initially studied in vitro during the induced DFs aging, both replicative and premature [70,110]):Morphological changes: the increase in size and flattening of the shape [111];Increased activity of senescence-associated lysosomal enzyme β-galactosidase (*SA-b-gal*) which is the “gold standard” for the identification of senCs both in vitro and in vivo (in tissue samples) [112];Increased levels of inhibitors of the cell cycle (p16INK4a, p21CIP1, and p53) [31,76];Visualization of cytoplasmic granularity under a light microscope: it indicates an increase in the number and size of lysosomes (while this does not mean an increase in the activity of these organelles since there is a marked decrease in the level of autophagy associated with lysosomes during aging) [88,93,113];Accumulation of lipofuscin [113];Increase in the frequency of γH2AX (a marker of double-stranded DNA breaks that occur during persistent DNA damage and DDR activation [76]);Presence of telomere-associated foci of DNA damage (TAF) [114];Decrease in the level of nuclear intermediate plate protein and epigenetic modulator of lamin B1 [115,116,117,118,119] (the level of lamin B1 decreases in vitro in senDFs regardless of the stress factor [117] and in vivo in DFs isolated from skin samples with signs of premature and chronological aging [120]);Presence of senescence-associated heterochromatin foci (SAHF, special heterochromatin structures formed in the nuclei of senCs) [121];Presence of DNA foci with chromatin changes that enhance cell aging (DNA-SCARS, DNA Segments with Chromatin Alterations Reinforcing Senescence) [69];Presence of HMGB1 (a protein from the group of nuclear non-histone proteins; in senCs it leaves the nucleus and moves to the cytoplasm and ECM; a decrease in its level in the nucleus leads to a decrease in gene expression) [122];High levels of proinflammatory cytokines in cells with SASP (in particular, IL-1α, IL-6, TNF, NF-κB, etc.), chemokines (CXCR2), metalloproteinases (MMP-3 and MMP-9), etc. [123,124,125];Mt dysfunction [84].

The DFs aging markers detected in vitro were also identified in vivo in the study of people with chronological aging [126,127] and photoaging [128]. Among these markers, the most common markers are the following:High level of SA-β-gal activity;Change in the production of ECM components;Increased level of cyclin-dependent kinases p21 and p16INK4a (among the other markers detected in vivo, it has the highest correlation with markers revealed in vitro [31]);Depletion of lamin B1 [117];Presence of SAHF (Senescent-Associated Heterochromatin Foci) [70,129];Increased level of SASP proinflammatory cytokines [124];Presence of telomere-associated foci of DNA damage (TAF) used as the quantitative marker of skin tissue aging in situ [114].

In addition, the high level of VEGF (vascular endothelial growth factor) can serve as one of the markers of senDFs in vivo, which is associated with increased vascular permeability, erythema, and the risk of skin cancer. It should be noted that the activation of VEGF production in senDFs does not depend on the type of cellular aging trigger [130]. In the scientific literature, much attention was paid to the identification of senDFs in the skin since there is no doubt about the key role of these cells in the induction and progression of skin-tissue aging [24,44,95,131,132]. According to Gorgoulis et al. (2019), for the most reliable identification of senCs in tissues (in situ), a “multi-marker approach” should be used, combining, for example, cytoplasmic markers (SA-β-gal and lipofuscin) with nuclear markers (p16^INK4A^ and p21^WAF1/Cip1^), as well as the markers associated with SASP [46]. The most frequently used methods are the following: single cell analysis, enzyme immunoassay, visualizing flow cytometry, or mass cytometry [46].

### 4.2. The Role of senDFs in Skin Aging

It has been revealed that, with age, an accumulation of senDFs occurs in the skin, which, through the associated specific SASP and by paracrine mechanisms, contributes to depletion of the SC pool, destruction/disruption of the regenerative abilities of the tissue, and, therefore, the induction and progression of its aging [14,44,132,133,134,135,136,137].

It should be noted that senCs belong to a specific type of cells. The results of preclinical studies have shown that selective removal of senCs significantly improves tissue homeostasis, prolongs the life of animals, and improves life quality [101,138]. However, it has also been shown that removal of senCs from the wound delays the healing, leads to fibrosis, and impairs the formation of granulation tissue [139]. In addition, it has been established that senCs participate in the remodeling/plasticity of body tissues [101]. These effects are related to the fact that cellular aging is an evolutionary antagonistic pleiotropic process while senCs are involved in every stage of ontogenesis: from embryogenesis, where they participate in the development of organs/tissues, to a mature state, where they play a significant role in the repair of organs/tissues by preventing the proliferation of cells with damaged DNA, thereby lowering the risk of neoplastic transformation [100,101,140].

In young people, the immune system effectively eliminates senCs from the body. However, the functions of the immune system decrease with age and the accumulation of senescent cells is observed in tissues [17]. As a result, the destruction of tissues occurs due to a number of pathophysiological processes mediated by SASP not only at the level of tissues but also at the level of cell populations.

At the level of tissues, degradation of the ECM is observed, caused by overproduction of MMPs in senescent cells [73]. Mild chronic aseptic inflammation also develops due to the secretion of many pro-inflammatory factors by senescent cells [14], including cytokines (IL-1, IL-6, and IL-8) and tumor necrosis factor (TNF), which recruit macrophages, neutrophils, and T cells infiltrating the tissues [100,141,142]. Thus, for example, it has been shown that IL-1 is directly associated with “paracrine” aging in vivo through activation of the inflammasome complex (a multi-protein oligomeric complex in myeloid cells responsible for activation of the inflammatory response) [135]. As a result, the following events occur: (1) degradation of membrane receptors, signaling pathways, proteins, and other components of the ECM; (2) changes in functions of the SCs niches; (3) disruption of autophagy processes; and (4) activation of transcription factor NF-κB (transcription nuclear factor of activated B cells), promoting the progression of inflammation in tissues [143]. The tissue inflammation is also accompanied by high levels of proinflammatory cytokines (IL-6, IL-1β, TGF-b, and TNF-α) that disrupt the transmission of anabolic signals, which leads to a decrease in the sensitivity of tissues to nutrients [144]. This close connection between aging and the inflammatory processes has been called “inflammaging” (inflammation + aging) or inflammatory aging.

At the level of cell populations, there are changes that are primarily associated with the paracrine mechanism, causing the aging phenotype in SCs located in spatial proximity to senCs [134,137,144,145,146,147,148], the so-called passive senescence [145] or “aging of an outsider observer” [146]. At the same time, the increase in the level of ROS is observed in senDFs, which causes the increase in Mt dysfunction. Thus, a vicious circle is formed (for more information, see above) that contributes to an even greater increase in ROS production and promotes the damaging effect of ROS on cell structures/organelles, which is accompanied by the reduced activity of antioxidant enzymes that is observed in cells with age [148,149]. In addition, factor NF-κB is activated and the “axis” ROS–NF-κB is formed, inducing DDR in SCs adjacent to senCs (the so-called “witness cells”) [44,48,94,147], which leads to the subsequent cell apoptosis or aging. At the same time, AP-1 and NF-B-dependent signaling pathways are also activated in senDFs, which contribute to the progression of inflammation in tissues [150].

Summarizing the abovementioned data and taking into account the main negative effects of senCs occurring in the skin [44,147], one can conclude the following about senescent cells:They do not proliferate, which leads to violation of the SCs’ self-renewing process and depletion of the SCs pool;Cause aging of neighboring SCs;Promote an increase in the level of ROS and cause Mt dysfunction;Induce DNA damage and aging of “witness cells” through the paracrine mechanism and ROS overproduction;Cause the chronic aseptic inflammation in tissues due to the effect of proinflammatory SASP factors secreted by senCs;Enhance the ECM degradation in the dermis by producing the high MMPs level;Disrupt cellular and tissue homeostasis.

Thus, the pathogenetic processes associated with the accumulation of dysfunctional senDFs in skin tissues with age lead to depletion of the SC pool and, therefore, to the decrease in regenerative abilities and the progression of aging processes in the tissues (Figure 5) [44,48,136,146,148]. The main research data confirming the role of senDFs in skin aging are the following [44,95,151]:An increase in the level of p16INK4a-positive DFs in the dermis correlating with the formation of wrinkles and the appearance of typical signs of elastic fiber aging;During chronological aging, DFs have a proteomic profile in situ identical to the senDFs profile;An increase in the ROS level leads to an increase in the number of p16INK4a-positive DFs in the skin and correlates with the progression of skin aging;The spread of senescence to neighboring DFs with the expression of characteristic markers of cellular aging was recorded during transplantation of human senDFs into the skin of young immunodeficient mice;Organ cultures obtained on the basis of human senDFs have signs of aging typical for chronological aging of the skin, including impairment of epidermal morphogenesis;DFs isolated from the skin of elderly people are characterized by a gene expression pattern similar to that of senDFs;Studies using a model of perforin-deficient mice (characterized by reduced functions of NK cells) have demonstrated the suppressed ability of the immune system to eliminate senDFs which by accumulating in the dermis lead to structural changes and the progression of aging processes in the dermis;The results of a clinical study conducted using the local application of rapamycin on the skin of elderly people (with chronological aging) showed a decrease in the level of p16INK4a-positive DFs, as well as a decrease in the number of fine wrinkles and an increase in the thickness and elasticity of the skin.Rapamycin is an inhibitor of mTOR (protein regulating the cell cycle and participating in the aging of DFs through the regulation of SASP) suppressing the translation of membrane-bound cytokine IL-1a and thereby inhibiting the secretion of pro-inflammatory SASP factors induced by IL-1a.

## 5. Conclusions

The considered molecular and cellular mechanisms observed in the skin during aging allow us to conclude that changes both in the SCs themselves and in their microenvironment (niches) occurring with age cause depletion of the SC population underlying the fibroblastic differon, which, in turn, leads to a decrease in the number of differentiated (mature) DFs responsible for ECM production and remodeling and, thus, to a decrease in the regenerative ability of the skin and the irreversible progression of skin aging.

## Figures and Tables

**Figure 1 ijms-23-06135-f001:**
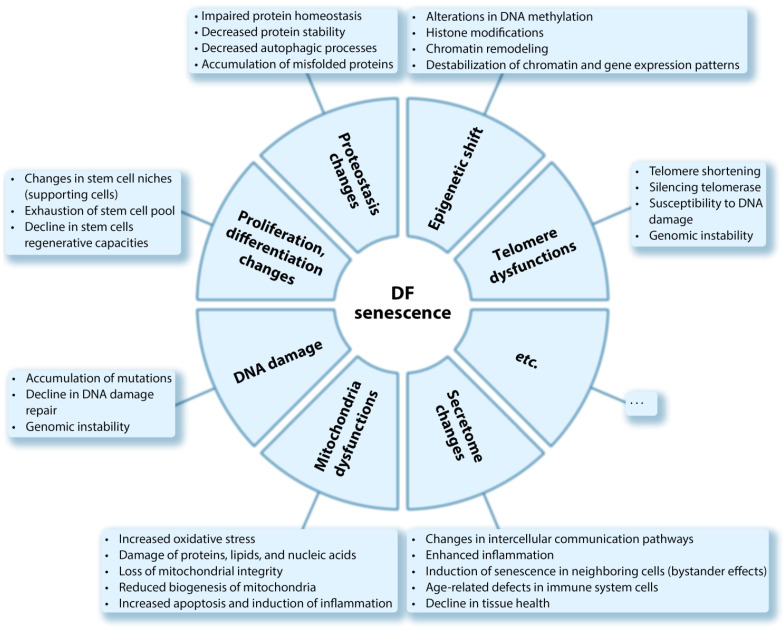
Characteristics of the cellular and molecular signs of cellular senescence.

**Figure 2 ijms-23-06135-f002:**
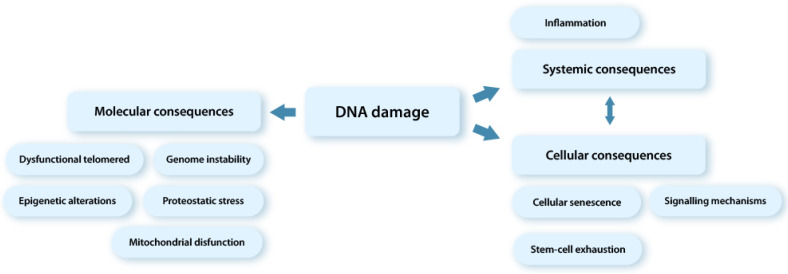
Consequences of DNA damage at the molecular, cellular, and tissue levels.

**Figure 3 ijms-23-06135-f003:**
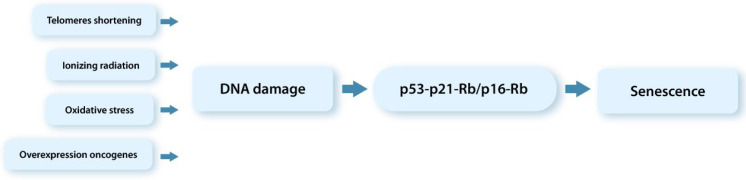
Schematic representation of cellular aging.

**Figure 4 ijms-23-06135-f004:**
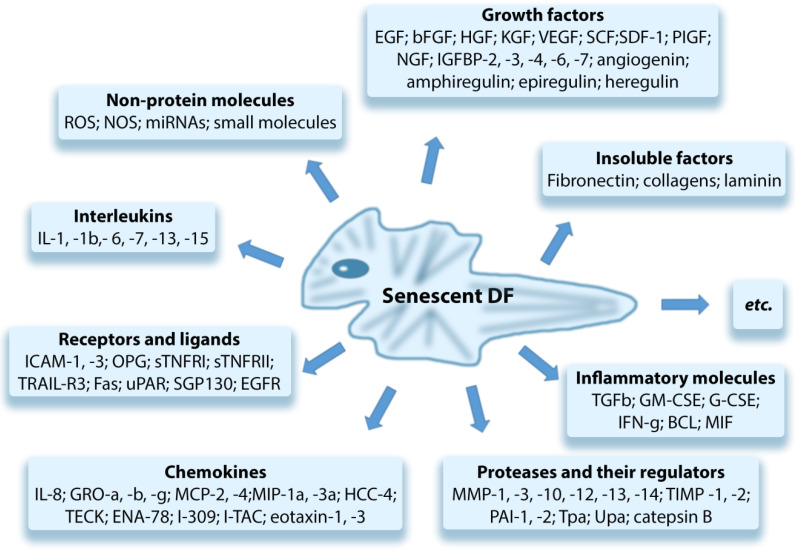
Components of the senescence-associated secretory phenotype (SASP).

**Figure 5 ijms-23-06135-f005:**
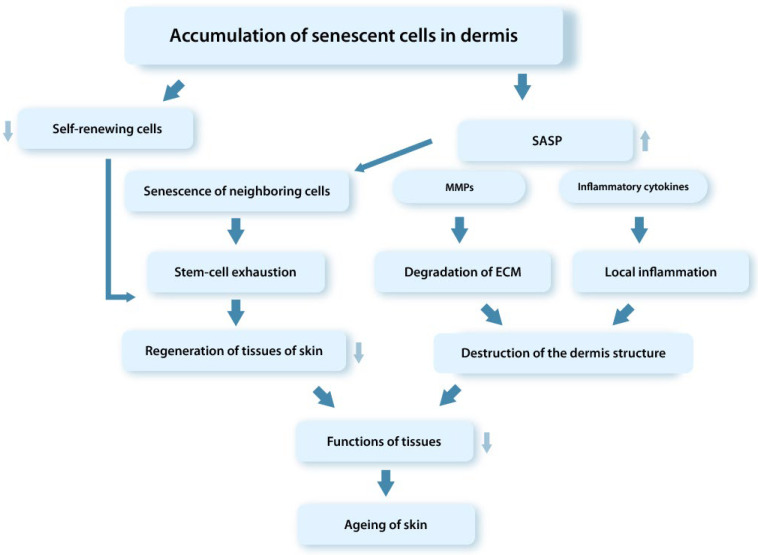
Schematic representation of mechanisms of the tissue dysfunction involving senCs (SASP). SenCs accumulating with age in tissues contribute through the SASP factors to the depletion of the SC pool, degradation of ECM in the dermis, development of the chronic aseptic inflammation in tissues, and disruption of the tissue regenerative potential.

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
