# Peer review of "Age-Related Changes in the Fibroblastic Differon of the Dermis: Role in Skin Aging"

_ijms, 2022, doi:10.3390/ijms23116135_

Round 1

Reviewer 1 Report

In the present manuscript, Zorina et al write a very interesting review about the Age-Related Changes in the Fibroblastic Differon of the Dermis: Role in skin ageing. 

it is a very comprehensive review that deals with several aspects of fibroblast loss of stemness that explain part of the events seen during skin ageing.

The revision has been well structured and deals with practical aspects such as the identification of senescent fibroblasts.

The review is structured in different sections that gather important aspects of the different changes experienced by fibroblasts during the process of ageing.

A revision of English can be conducted although English seems to be correct and proper.

As a minor comment, the format of the tables seems to be a bit difficult to follow and not very attractive. The text of the tables could be reduced or schematised, so the information can be better seen?

Author Response

As a minor comment, the format of the tables seems to be a bit difficult to follow and not very attractive. The text of the tables could be reduced or schematised, so the information can be better seen?

We are grateful to the reviewer for evaluating our manuscript and, according to his wishes; we converted the tables into figures or simply text. We hope this has improved the overall look of our work.

Reviewer 2 Report

the manuscript is intersting and contains useful data. I have several suggestion to improve the content^

1) It would be nice to clarify what cells are meant by the term "dermal stem cells" - whether thay are mesenchymal stem cells (pericytes, skin progenitors, etc), hair follicle stem cells, stem cells from adipose tissue or other types. Examples on their depletion should contain more detailed information concerning the type of cells and their relation to the dermis.

2) I am not sure that presentation of tables and schemes from other papers, even with modifications, is a good idea. Try to create your own ones, on the basis of several papers and overall review.

Figure 3 - "dermis", not "derma"

Author Response

the manuscript is interesting and contains useful data. I have several suggestion to improve the content:

We are grateful to the reviewer for evaluating our manuscript and making corrections according to all his wishes.

1) It would be nice to clarify what cells are meant by the term "dermal stem cells" - whether thay are mesenchymal stem cells (pericytes, skin progenitors, etc), hair follicle stem cells, stem cells from adipose tissue or other types. Examples on their depletion should contain more detailed information concerning the type of cells and their relation to the dermis.

Explanations have been added to the text: "This is probably due to the predominance of stem/progenitor cells (SCs, tissue-specific mesenchymal cells, precursors of fibroblasts of the skin (outside the hair follicles and mesenchymal stem cells, like bone marrow stem cells and others that were not incorporated into the skin during embryonic development) in the composition of papillary fibroblast populations in which changes occur primarily."

2) I am not sure that presentation of tables and schemes from other papers, even with modifications, is a good idea. Try to create your own ones, on the basis of several papers and overall review.

Tables and figures have been revised; references are left in the text only.

3) Figure 3 - "dermis", not "derma"

"dermis" instead  "derma" corrected, now it is in fig.5.

We hope that the revised manuscript is acceptable for publication.